# Analysis of BC Pollution Characteristics under PM$_{2.5}$ and O$_3$ Pollution Conditions in Nanjing from 2015 to 2020

**Yuxuan Pei [1], Honglei Wang [1],\*, Yue Tan [2], Bin Zhu [1], Tianliang Zhao [1], Wen Lu [1] and Shuangshuang Shi [1]**

[1] Key Laboratory for Aerosol-Cloud-Precipitation of China Meteorological Administration, Nanjing University of Information Science and Technology, Nanjing 210044, China

[2] Department of Civil and Environmental Engineering, Hong Kong Polytechnic University, Hong Kong 999077, China

\* Correspondence: hongleiwang@nuist.edu.cn

**Abstract:** Using an AE-33 Aethalometer, surface air pollution monitoring data, radiosonde data, and conventional meteorological observation data, the characteristics and influencing factors of black carbon (BC) pollution under PM$_{2.5}$ and O$_3$ pollution in Nanjing were comprehensively analyzed. The results show that the air quality saw an apparent trend of improvement from 2015 to 2020, and the number of days with excellent air quality increased by 38.2% from 2015 to 2020. The number of days when the dominant pollutant was PM$_{2.5}$ decreased each year to only 18 days in 2020, with an annual rate of decline of 16.0% from 2015 to 2020. The number of days when the dominant pollutant was O$_3$ increased, reaching a maximum for the 6-year period of 78 days in 2019, with an annual rate of increase of 11.1% in 2015–2019. The average mass concentration of BC when the dominant pollutant was PM$_{2.5}$ in slight, moderate, and heavy pollution decreased in 2015–2016 and then showed an increasing yearly trend in 2016–2018, with annual rates of increase of 73.8%, 105.5%, and 156.3%, respectively, reaching a maximum in 2018 and then starting to decrease thereafter. With PM$_{2.5}$, the slight pollution and moderate pollution BC mass concentrations were mainly influenced by the height of the inversion layer. The average BC mass concentrations in the case of slight and moderate pollution with O$_3$ as the dominant pollutant decreased significantly from 2015 to 2016, and then increased yearly from 2016 to 2019, with annual rates of increase of 112.2% and 138.6%, respectively, reaching a maximum in 2019 and then decreasing from 2020. The BC mass concentration was significantly negatively correlated with wind speed in both light and moderate O$_3$ pollution, with correlation coefficients of −0.79 and −0.68, respectively. The seasonal variation and dominant influencing factors of BC differed when PM$_{2.5}$ and O$_3$ were the dominant pollutants. When PM$_{2.5}$ was the dominant pollutant, the seasonal variation in the BC for slight pollution was winter > autumn > summer > spring, and for moderate pollution and heavy pollution was autumn > winter > spring, which were mainly affected by the inversion stratification difference and wind speed. When O$_3$ was the dominant pollutant, the seasonal variation in BC under slight pollution was autumn > summer > spring, and for moderate pollution, it was spring > summer > autumn, which were mainly affected by the wind speed. Studying the evolution of BC in air pollution under different dominant pollutants is important to further improve the capability and level of global climate change research and predictions and can provide a scientific basis for assessing their impact on the environment, health, and climate.

**Keywords:** Nanjing; black carbon; annual and seasonal variations; meteorological elements; boundary layer structure

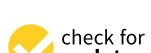



## 1. Introduction

Black carbon (BC) is the most important absorbing aerosol in the atmosphere. Although it only accounts for 5–15% of the mass concentration of atmospheric aerosols [1–3], its radiative properties can significantly influence the global radiation balance [4–8] and the evolutionary development of the urban boundary layer [7,8]. It is produced by the

incomplete combustion of carbon-containing materials, of which there are two main types: fossil fuel combustion and biomass combustion [9]. It has a short residence time in the atmosphere, usually a few days to two weeks [10], and absorbs solar radiation while releasing infrared radiation, thus heating the atmosphere and causing the greenhouse effect. It has been suggested that the warming effect of BC is about two-thirds of that of $CO_2$ [11] and that reducing BC emissions may help to mitigate some of the atmospheric warming caused by their increase [12]. During atmospheric transport, the surface of BC can adsorb other pollutants (e.g., $SO_2$, $O_3$), providing an active carrier for the gas-particle conversion process of these substances and acting as a catalyst [13]. Its optical absorption properties in cities with high BC mass concentrations can drastically reduce atmospheric visibility and significantly degrade air quality. BC deposited on ice and snow surfaces has been proven to reduce surface albedo by absorbing more solar radiation, which may accelerate glacier melting and result in changes in glaciers [14,15]. BC is also a huge health hazard as tiny BC particles can enter the human respiratory system and lungs [16]. In addition, BC can affect plant growth and the ecosystem [17].

Since the 21st century, with the rapid economic development of China and the dramatic increase in population and energy consumption, China has entered a composite pollution stage where aerosols and $O_3$ coexisted. In contrast, many scholars have conducted plenty of studies on the characteristics of air pollution and the complex mechanisms of its generation under this complex pollution. Pu et al. [18] analyzed BC evolution at the WMO/GAW Station at Mt. Waliguan from 1994 to 2017 and the results showed that the annual average BC concentration from 2001 to 2012 ranged from $1.9 \times 10^2$ ng·m$^{-3}$ to $5.1 \times 10^2$ ng·m$^{-3}$ with a growth rate of 29%. However, from 2012 to 2016, BC concentrations decreased at a rate of 64%. In addition, BC transport was found to be closely related to wind transport. Huang et al. [19] analyzed historical monitoring data of the air quality index (AQI) in the Pearl River Delta from 2015 to 2019 and the results showed that the trend in the annual average AQI in the Pearl River Delta was consistent. The annual average AQI decreased and then increased with a W-shaped change curve and in autumn, the ozone concentration increased sharply and became the main pollutant in ambient air. Shen et al. [20] analyzed the regional transport patterns of air pollutants from Central and Eastern China (CEC) to the THB driven by strong cold airflows and the results showed that passages of strong, cold airflow could quickly purge air pollutants over the source region, e.g., the North China Plain (NCP), whereas the regional transport of air pollutants from the source region could exacerbate air quality in the receptor region of the Twain-Hu Basin (THB) due to the cold airflows modulated by atmospheric circulations over China. Zhang et al. [21] analyzed the evolution of atmospheric $PM_{2.5}$ and $O_3$ pollution in Beijing from 2013 to 2020 and the results showed that the frequency, duration, and peak concentration of the $PM_{2.5}$ pollution processes decreased yearly from 2013 to 2020; in contrast, the interannual variation trend in the $O_3$ pollution processes was not evident. Guo et al. [22] analyzed the characteristics of the air quality changes in Nanjing from 2015 to 2021. The results showed a decreasing trend in the number of exceedance days of $NO_2$, $PM_{10}$, and $PM_{2.5}$ in Nanjing from 2015 to 2020, with an average decrease of 29.1%, 38.1%, and 28.1%, whereas the number of exceedance days of $O_3$ showed an increasing trend. Zhang et al. [23] conducted a BC study in Hefei, a city in central China, and analyzed the seasonal, daily, and monthly variation characteristics of BC, showing that the annual mean BC concentration was $3.5 \pm 2.5$ μg·m$^{-3}$ and the seasonal aspects of the BC mass concentration were the lowest in summer, medium in autumn and spring, and the highest in winter. Cao et al. [24] analyzed the concentration distribution characteristics of the carbon fraction in $PM_{2.5}$ in 14 Chinese cities in winter and summer. The results showed that the concentrations of both OC and EC were the highest in winter and lower in summer in 14 cities, on the one hand, due to more carbon aerosol emissions from coal heating in winter, and on the other hand, due to the lower height of the inversion layer in winter, which makes the pollutants less likely to disperse. Su et al. [25] focused on the role of aerosols in thermodynamic stability and PBL development by studying vertical distribution. It was found that the aerosol–PBL

interactions could be enhanced by anti-aerosol structures or potentially neutralized by reduced structures. Furthermore, aerosols can both enhance or inhibit PBL stability, leading to positive or negative feedback loops. Quan et al. [26] compared the boundary layer heights and the corresponding pollutant concentrations under different weather conditions. They showed that the boundary layer heights and the near-ground pollutant concentrations were negatively correlated on both clean and polluted days. Petaja et al. [27] found that the feedback mechanism between the boundary layer height and pollutant concentration was weak when the $PM_{2.5}$ concentration was below 200 $\mu g \cdot m^{-3}$ and that in heavy pollution, a large number of pollutant emissions caused the boundary layer height to decrease, resulting in a further accumulation of pollutants.

As one of the fastest growing economic regions in the world, the Yangtze River Delta region has high BC emissions, which significantly impact regional climate and air quality [28]. Nanjing, the central city of the Yangtze River Delta and East China, has the characteristics of compound pollution in the atmosphere. BC pollution in Nanjing is relatively heavy and various sources of BC have been found [29]. However, in the context of composite atmospheric pollution, few studies divide air pollution into the different dominant pollutants and then further explore the characteristics of BC pollution under a long time series, or examine the relationship between the characteristics of the BC aerosol changes and boundary layer characteristics in different years and seasons, as well as the differences in the influence of their meteorological elements, which indicate that further analysis and exploration is needed. With this as the starting point, this paper first applies the Ambient Air Quality Standard (GB3095-2012) to classify the air pollution in the Nanjing area from 2015 to 2020 into air pollution with $PM_{2.5}$ and $O_3$ as the dominant pollutants. Then, a BC pollution characterization was performed using the BC mass concentration observed by a multi-wavelength Aethalometer (AE-33). The atmospheric boundary layer sounding data and meteorological element characteristics of the selected corresponding period are also used to explore the influence on the BC pollution characteristics. Thus, the BC pollution characteristics under different dominant pollutants and their influencing factors are derived, which are important for further improving the capability and level of global climate change research and predictions and can provide a scientific basis for strategies for regional air pollution prevention and control, as well as an assessment of the impact on the environment, health, and climate.

## 2. Experiment and Methods

### 2.1. Site Introduction

The observation site is located in Nanjing, the capital of Jiangsu Province (32.2° N, 118.7° E). The area is located in the westerly wind belt and has a typical monsoonal climate, with northerly winds predominating in winter and spring and westerly winds in summer and autumn. The specific location was chosen as the BC sampling site on the roof of the meteorological building of the Nanjing University of Information Engineering, which is about 40 m above ground level and has an average altitude of about 62 m. The Ningliu Road, with six lanes in both directions, is about 500 m to the east of the site; the Longwang Mountain scenic area is about 900 m to the southeast and its altitude is about 100 m. The observation site is located in the eastern part of the campus of the Nanjing University of Information Engineering, and there are primarily residential areas to the north. About 6 km southeast of the university is the Nanjing Industrial Park, where various large chemical and steel enterprises, such as Yangzi Petrochemical and the Nangang Group, are located. The site is located in the central area of the Yangtze River Delta, which is very conducive to promoting systematic observations and research on BC aerosols in the Yangtze River Delta region and is an ideal observation site.

### 2.2. Observation Instruments and Data

Near real-time continuous measurements of BC mass concentrations were carried out using an Aethalometer (Model AE-33 from Magee Scientific, Berkeley, CA, USA), which

can revise and reduce the effects of loading through a two-point measurement technique, making the instrument's measurement performance more efficient and accurate [30]. In this instrument, atmospheric air is pumped through the inlet at the required flow rate of 5.0 L·min$^{-1}$ and a sampling cutting joint for PM$_{2.5}$ is connected. Moreover, the instrument has seven measurement channels at 370, 470, 525, 590, 660, 880, and 940 nm. The black carbon meter uses optical greyscale measurements to infer the mass concentration of BC by measuring the attenuation of BC on a quartz filter membrane to invert the optical absorption coefficient of BC. The AE-33 Black Carbon Meter has an accuracy of 1 ng·m$^{-3}$ and a sampling frequency of 5 min, and the subsequent analysis processes the data into 1 h averages, from which the daily and monthly averages of the BC mass concentrations are then obtained.

The air quality levels used in this paper were obtained from historical data from China's air quality online monitoring and analysis platform. Available online: https://www.aqistudy.cn/historydata/ (accessed on 9 August 2021). According to the "Ambient Air Quality Standards" (GB3095-2012), issued by the former Ministry of Environmental Protection in 2012, a polluted day with PM$_{2.5}$ as the dominant pollutant and a polluted day with O$_3$ as the dominant pollutant is defined as a day when the 24 h average mass concentration of PM$_{2.5}$ is greater than the national air quality secondary concentration limit (75 μg·m$^{-3}$). An eighth average mass concentration of O$_3$ is greater than the O$_3$ national air quality second-level concentration limit (160 μg·m$^{-3}$). Meteorological elements data were obtained from the atmospheric sounding base at the Nanjing University of Information Engineering and included temperature, pressure, humidity, wind, and visibility, with a temporal resolution of 1 h. The atmospheric boundary layer data were obtained from daily soundings at 08:00 and 20:00 from the University of Wyoming weather data website. Available online: http://www.weather.uwyo.edu/upperair/sounding.html (accessed on 9 August 2021).

## 3. Results and Discussion

### 3.1. Characteristics of Air Quality Class Changes

In Figure 1a, it can be seen that the annual variation in the number of days with different air quality classes during the observation period varied significantly. The corresponding PM$_{2.5}$ and O$_3$ concentration limits of the air quality classes are referred to in Table 1. The number of days with excellent air quality in 2015–2020 showed a significant upward trend, from 33 days in 2015 to 96 days in 2020, with an annual increase of 38.2%; the number of days with good air quality did not change significantly, all within 210 days; the number of days with slight air quality pollution showed a decreasing trend overall, but the number of days with slight pollution increased in 2019. The number of days with moderate air quality pollution decreased yearly from 2015 to 2020, from 25 days in 2015 to only six days in 2020, with an annual rate of decline of 15.2%; the number of days with heavy pollution showed a relatively apparent downward trend, with no heavy pollution in 2020; and there was only one day with serious pollution in 2017. It can be seen that the air quality from 2015 to 2020 showed a clear trend of improvement.

**Table 1.** Air quality levels corresponding to PM$_{2.5}$ and O$_3$ concentration limits.

| Air Quality Levels | PM$_{2.5}$ 24 h Average (μg·m$^{-3}$) | O$_3$ 8 h Sliding Average (μg·m$^{-3}$) |
|---|---|---|
| Excellent | $0 < PM_{2.5} \leq 35$ | $0 < O_3 \leq 100$ |
| Good | $35 < PM_{2.5} \leq 75$ | $100 < O_3 \leq 160$ |
| Slight pollution | $75 < PM_{2.5} \leq 115$ | $160 < O_3 \leq 215$ |
| Moderate pollution | $115 < PM_{2.5} \leq 150$ | $215 < O_3 \leq 265$ |
| Heavy pollution | $150 < PM_{2.5} \leq 250$ | $265 < O_3 \leq 800$ |
| Serious pollution | $250 < PM_{2.5}$ | $800 < O_3$ |

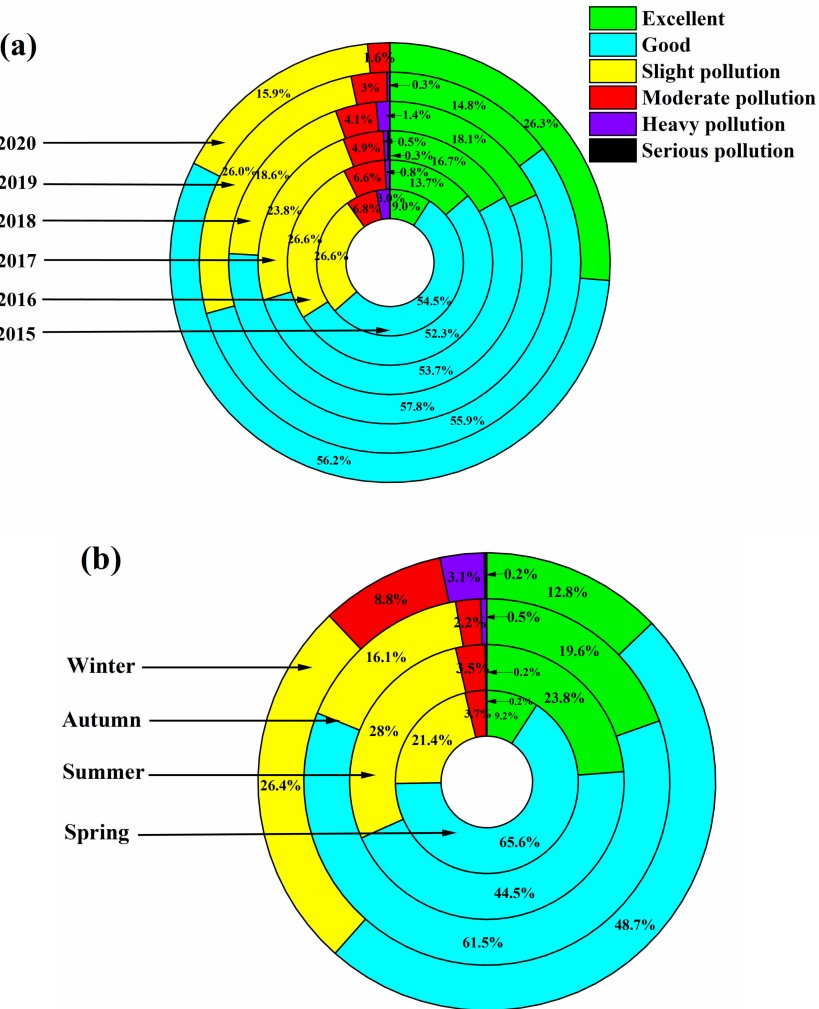

**Figure 1.** (**a**) Annual proportions of days with different air quality levels in 2015–2020; (**b**) Seasonal proportions of days with different air quality levels in 2015–2020.

As shown in Figure 1b, the seasonal differences in the number of days with different air quality levels during the observation period were significant. The proportion of days with slight pollution was summer (28.0%) > winter (26.4%) > spring (21.4%) > autumn (16.1%); the proportion of moderate pollution was 8.8% in winter, which was 2.3, 2.5, and 4.0 times higher than those in spring, summer, and autumn respectively; the proportion of days with heavy pollution was 3.1% in winter, which was significantly higher than those in spring, summer, and autumn. In addition, there was one day of serious pollution in winter. As can be seen, slight pollution was most likely to occur in summer, whereas moderate, heavy, and serious pollution were all concentrated in winter.

In Figure 2a,b, it can be seen that the days with $PM_{2.5}$ and $O_3$ as the dominant pollutants in Nanjing from 2015 to 2020 were 268 and 319, respectively. The number of days with $PM_{2.5}$ and $O_3$ as the dominant pollutants accounted for 94.2% of all polluted days, indicating that $PM_{2.5}$ and $O_3$ pollution were the dominant air pollutants in Nanjing.

In Figure 2c, it can be seen that when the dominant pollutant was $PM_{2.5}$, the number of $PM_{2.5}$ pollution days was 39, 3, 41, and 185 in spring, summer, autumn, and winter, respectively. The number of pollution days in winter was significantly higher than in the other seasons, being 374.4%, 606.7%, and 351.2% higher than in spring, summer, and autumn, respectively. The number of moderately polluted and heavily polluted days in winter was 47 and 16, respectively, which was also significantly higher than in the other seasons, in addition to one day of serious pollution in winter. This means that $PM_{2.5}$ pollution was mainly concentrated in winter and was more serious, mainly owing

to the high rainfall and wet solid deposition in summer, which resulted in lower mass concentrations of atmospheric pollutants. In contrast, the lower boundary layer and stable stratification in winter were more likely to cause $PM_{2.5}$ accumulation.

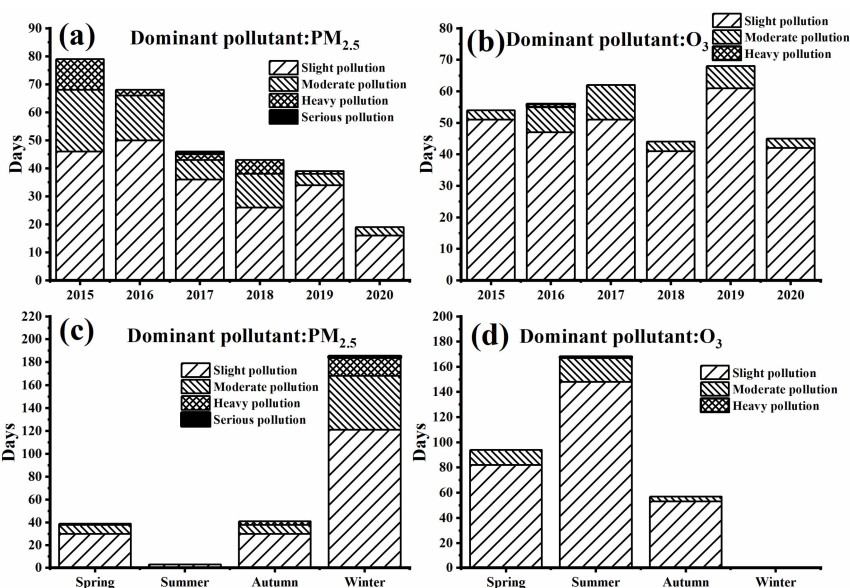

**Figure 2.** (**a**) Annual variations in polluted days for the dominant pollutant $PM_{2.5}$ in 2015–2020; (**b**) Annual variations in polluted days for the dominant pollutant $O_3$ in 2015–2020; (**c**) Seasonal variations in polluted days for the dominant pollutant $PM_{2.5}$ in 2015–2020; (**d**) Seasonal variations in polluted days for the dominant pollutant $O_3$ in 2015–2020. From Figure 2a, it can be seen that the number of days with $PM_{2.5}$ as the dominant pollutant decreased yearly, with 89 in 2015 and only 18 in 2020, with an annual rate of decrease of 16.0%. Meanwhile, the number of slightly polluted, moderately polluted, and heavily polluted days also showed a significant decline. The number of days with slight pollution and moderate pollution in 2015 were 46 and 32, respectively, whereas in 2020, there were only 15 and 3 days that, when compared with the number of days in 2015, fell by 67.4% and 90.6% respectively, and there was no heavy pollution in 2020. This is due to the implementation of measures to adjust the industrial, energy, and transport structures in the "Action Plan for the Prevention and Control of Air Pollution" and the "Three-Year Action Plan for the Defense of the Blue Sky implemented from 2018 to 2020". Available online: http://zrzy.tianshui.gov.cn/dayin-c9c80aba59ed471fa05f7eddd55abeea.htm (accessed on 30 August 2021). These actions have significantly reduced the total emissions of major air pollutants, resulting in a significant decrease in $PM_{2.5}$ and significantly reducing the number of $PM_{2.5}$ pollution days. In Figure 2b, it can be seen that when the dominant pollutant was $O_3$, the number of polluted days was predominantly slight, with an average annual percentage of 87.3%. The number of days with $O_3$ as the dominant pollutant increased from 54 in 2015 to 78 in 2019, which was the highest number of days in six years, with an annual increase of 11.1%. This may be due to the decrease in the number of days polluted by $PM_{2.5}$, which reduced the aerosol optical thickness, increased the amount of light reaching near the ground, and accelerated the photolytic reaction rate, thus leading to an increase in $O_3$ pollution. It also indicated that the dominant pollutant of air pollution in Nanjing gradually changed from $PM_{2.5}$ to $O_3$.

In Figure 2d, it can be seen that when the dominant pollutant was $O_3$, the number of days with $O_3$ pollution was 94, 168, and 57 in spring, summer, and autumn, respectively. The number of polluted days was higher in summer than in the other seasons, being 78.7% and 194.7% higher than in spring and autumn, respectively. At the same time, there were no days with $O_3$ as the dominant pollutant in winter. The number of moderately polluted days was 12, 19, and 4 in spring, summer, and autumn, respectively, indicating that $O_3$ pollution was mainly concentrated in summer and was more likely to occur at higher concentrations.

This is mainly due to the stronger solar radiation and longer days of light hours in summer, which intensifies the photochemical reactions and is more conducive to $O_3$ production.

In addition, Figure 2 shows that the total number of days with moderate, heavy, and serious pollution was 31.3% and 11.3% for $PM_{2.5}$ and $O_3$, respectively, indicating that pollution in Nanjing tends to be heavier when the dominant pollutant is $PM_{2.5}$.

### 3.2. Characteristics of BC Mass Concentration Changes under Different Air Quality Classes

In Figure 3a,b, it can be seen that when the air quality was under slight pollution, the BC mass concentration decreased in 2015–2016 when the dominant pollutant was $PM_{2.5}$, followed by a yearly upward trend in 2016–2018, with an annual rate of increase of 73.8%, reaching a maximum value of 5.6 $\mu g \cdot m^{-3}$ in 2018, followed by a yearly downward trend during 2018–2020, with an annual rate of decline of 9.6%; the BC mass concentration decreased significantly in 2015–2016 when the dominant pollutant was $O_3$, being 70.6% lower in 2016 than in 2015, then increased significantly in 2016–2019, reaching a maximum value of 4.0 $\mu g \cdot m^{-3}$ in 2019, with an annual rate of increase of 112.2%; and finally, the BC mass concentration in 2019–2020 decreased in mid-2019, with 2020 being 25.0% lower than 2019. In Figure 4b, it can be seen that the correlation coefficient between the wind speed and BC in 2015–2020 was −0.79, which was highly negatively correlated, and then in Figure 4f, it can be seen that there was no significant correlation between the BC mass concentration and visibility, indicating that the dominant pollutant was $O_3$ mainly when the lower wind speed caused BC to be less diffusible, thus increasing the BC mass concentration. It can be seen that the BC mass concentrations were lower in 2020 than in 2018 and 2019 when the dominant pollutant was either $PM_{2.5}$ or $O_3$, which is related to the reduction in anthropogenic emissions due to measures, such as industrial shutdowns and traffic closures, during COVID-19.

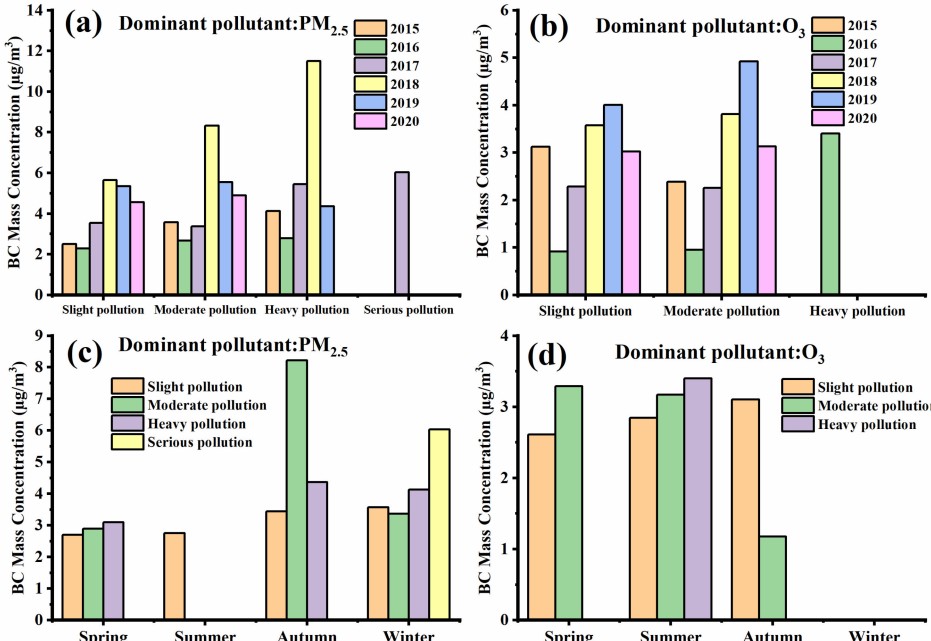

**Figure 3.** (**a**) Annual variations in BC mass concentrations at different air quality levels for the dominant pollutant $PM_{2.5}$ in 2015–2020; (**b**) Annual variations in BC mass concentrations at different air quality levels for the dominant pollutant $O_3$ in 2015–2020; (**c**) Seasonal variations in BC mass concentrations at different air quality levels for the dominant pollutant $PM_{2.5}$ in 2015–2020; (**d**) Seasonal variations in BC mass concentrations at different air quality levels for the dominant pollutant $O_3$ in 2015–2020.

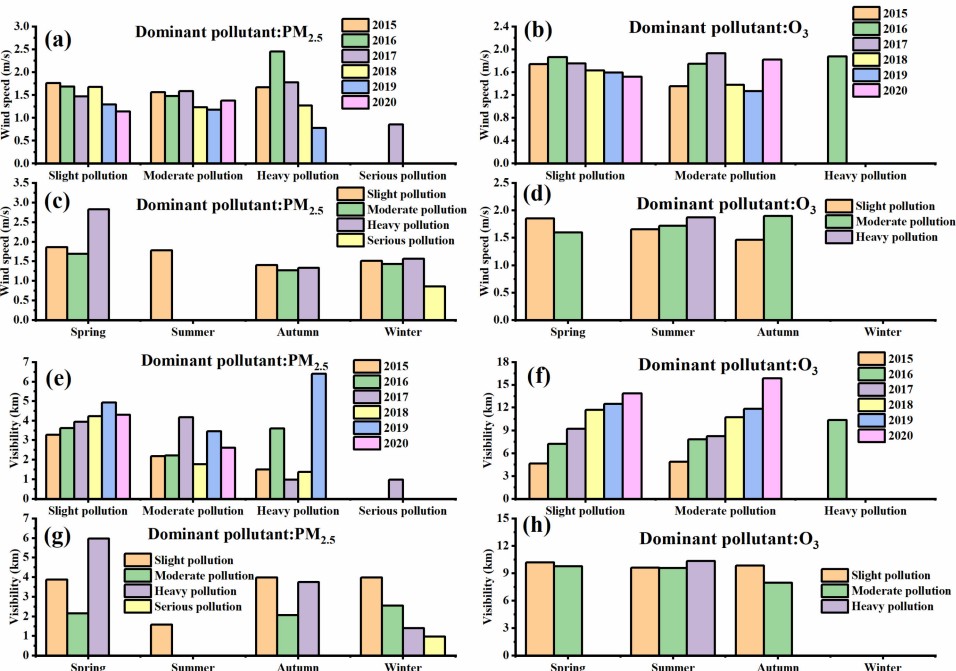

**Figure 4.** (**a**) Annual variations in wind speed at different air quality levels for the dominant pollutant PM$_{2.5}$ in 2015–2020; (**b**) Annual variations in wind speed at different air quality levels for the dominant pollutant O$_3$ in 2015–2020; (**c**) Seasonal variations in wind speed at different air quality levels for the dominant pollutant PM$_{2.5}$ in 2015–2020; (**d**) Seasonal variations in wind speed at different air quality levels for the dominant pollutant O$_3$ in 2015–2020; (**e**) Annual variations in visibility at different air quality levels for the dominant pollutant PM$_{2.5}$ in 2015–2020; (**f**) Annual variations in visibility at different air quality levels for the dominant pollutant O$_3$ in 2015–2020; (**g**) Seasonal variations in visibility at different air quality levels for the dominant pollutant PM$_{2.5}$ in 2015–2020; (**h**) Seasonal variations in visibility at different air quality levels for the dominant pollutant O$_3$ in 2015–2020.

In Figure 3a,b, it can be seen that when the air quality was under moderate pollution, the annual change in BC when the dominant pollutant was PM$_{2.5}$ was similar to that under slight pollution, with mass concentrations both decreasing in mid-2015–2016 and then both showing a year-on-year increase from 2016 to 2018, with an annual increase of 105.5%, peaking at 8.3 $\mu$g·m$^{-3}$ in 2018, which was significantly higher than those in other years. As can be seen in Figure 4e, visibility also reached a minimum value of 1.8 km in 2018, indicating that the high concentration of PM$_{2.5}$ in 2018 was the leading cause of the occurrence of high BC concentrations, with a decreasing trend in all BC from 2018 to 2020 and an annual rate of decrease of 20.6%; the annual variation under moderate pollution of BC when the dominant pollutant was O$_3$ was similar to that under slight pollution, with the mass concentrations both decreasing significantly in 2015–2016, being 60.0% lower in 2016 than in 2015, then showing a yearly increasing trend in 2016–2019, reaching a maximum value of 4.9 $\mu$g·m$^{-3}$ in 2019, with an annual rate of increase of 138.6%; and finally, both decreased in 2019–2020. In Figure 4b, it can be seen that the BC mass concentration changes in 2015–2020 were negatively correlated with wind speed to some extent, with a correlation coefficient of −0.68. It can be seen in Figure 4f that the correlation between the BC mass concentration and visibility was not significant, which is mainly due to the lower wind speed, which tends to make BC accumulate and thus the concentration increases.

In Figure 3a, it can be seen that when the air quality was under heavy pollution, the annual change in the BC mass concentration when the dominant pollutant was PM$_{2.5}$ was similar to that of slight and moderate pollution, both decreasing in the middle of 2015–2016 and then increasing year by year from 2016 to 2018, reaching a peak of 11.5 $\mu$g·m$^{-3}$ in 2018, with an annual rate of increase of 156.3%. As shown in Figure 4a,e, this is related to the relatively lower wind speed and higher concentration of PM$_{2.5}$ in 2018, which caused the

BC mass concentration to rise sharply, followed by a significant downward trend in 2018, being 62.1% lower in 2019 than in 2018.

In Figure 3c,d, it can be seen that when the air quality was under slight pollution, the seasonal variation in the BC mass concentration was winter (3.6 $\mu g \cdot m^{-3}$) > autumn (3.4 $\mu g \cdot m^{-3}$) > summer (2.8 $\mu g \cdot m^{-3}$) > spring (2.7 $\mu g \cdot m^{-3}$) when the dominant pollutant was $PM_{2.5}$. Figure 4c shows that the relatively lower wind speed in autumn and winter under slight pollution led to relatively higher BC mass concentration. The seasonal variation in the BC mass concentration was autumn (3.1 $\mu g \cdot m^{-3}$) > summer (2.8 $\mu g \cdot m^{-3}$) > spring (2.6 $\mu g \cdot m^{-3}$) when the dominant pollutant was $O_3$, and the seasonal variation in the wind speed was spring (1.9 $m \cdot s^{-1}$) > summer (1.7 $m \cdot s^{-1}$) > autumn (1.5 $m \cdot s^{-1}$), which can be seen in Figure 4d, indicating that the lower wind speed under slight pollution makes BC less likely to disperse in autumn, resulting in higher concentrations.

In Figure 3c,d, it can be seen that when the air quality was under moderate pollution, the BC mass concentrations were 2.9 $\mu g \cdot m^{-3}$, 8.2 $\mu g \cdot m^{-3}$, and 3.4 $\mu g \cdot m^{-3}$ in spring, autumn, and winter, respectively, when the dominant pollutant was $PM_{2.5}$ and were significantly greater in autumn than in spring and winter, being 182.8% and 141.2% higher, respectively. In Figure 4c, it can be seen that the seasonal variation in the wind speed under moderate pollution was spring (1.7 $m \cdot s^{-1}$) > winter (1.4 $m \cdot s^{-1}$) > autumn (1.3 $m \cdot s^{-1}$); the relatively lower wind speed in autumn makes it difficult to diffuse BC so the concentration increase. In addition, the average daily precipitation values during this period were 1.91 mm, 0.53 mm, and 0.89 mm in spring, autumn, and winter, respectively, calculated from the date of the observation period. The average daily precipitation values in autumn were 260.4% and 67.9% lower than those in spring and winter, respectively, so the wet removal effect in autumn during this period was smaller, which led to a further increase in the BC mass concentration. (63.6% and 62.5%, respectively). In Figure 4d, it can be seen that the seasonal variation in the wind speed was autumn (1.9 $m \cdot s^{-1}$) > summer (1.7 $m \cdot s^{-1}$) > spring (1.6 $m \cdot s^{-1}$), and the greater wind speed in autumn made BC easier to disperse and the concentration decreased.

In Figure 3c, it can be seen that when the air quality was under heavy pollution, the seasonal change in the BC mass concentration when the dominant pollutant was $PM_{2.5}$ was autumn (4.4 $\mu g \cdot m^{-3}$) > winter (4.1 $\mu g \cdot m^{-3}$) > spring (3.1 $\mu g \cdot m^{-3}$). Figure 4c shows that the seasonal variation in the wind speed under heavy pollution was spring (2.8 $m \cdot s^{-1}$) > winter (1.6 $m \cdot s^{-1}$) > autumn (1.3 $m \cdot s^{-1}$), which indicates that the lower wind speed in autumn tends to accumulate pollutants during heavy pollution, resulting in high BC mass concentrations.

*3.3. Influence of Boundary Layer Characteristics on BC Mass Concentrations at Different Air Quality Levels*

The inverse thermocline within the boundary layer is similar to a "dome," which prevents the diffusion of pollutants to higher altitudes, causing a significant accumulation of pollutants in the area below the inverse thermocline [7]. In Figure 5c,d, it can be seen that when the air quality was under slight pollution, the daytime BC quality concentrations showed an overall increasing trend from 2016 to 2019, with annual rates of increase of 41.2% and 116.9% respectively, regardless of whether the dominant pollutant was $PM_{2.5}$ or $O_3$. In Figure 5, it can be seen that BC was significantly negatively correlated with the daytime inverse thermosphere height in 2016–2019, with the correlation coefficients calculated to be −0.70 and −0.93, respectively, indicating that a lower inverse thermosphere height was more likely to confine BC to the near ground layer, making it less likely to disperse, resulting in higher BC mass concentrations. It can also be seen that when the dominant pollutant was $PM_{2.5}$, the inversion thickness in the daytime in 2018 and 2019 was 256.5 m and 234.1 m, respectively, which is significantly higher than that in other years, which is also the reason for the high values of the BC mass concentration in 2018 and 2019. When the dominant pollutant was $O_3$, the inversion thickness and inversion intensity in the daytime were not different, with an average of 89.3 m and 0.7 °C/100 m, respectively, and there was no significant correlation with BC. In Figure 6c,d, it can be seen that when the air quality was under slight pollution, whether the dominant pollutant was $PM_{2.5}$ or $O_3$, the

annual change in the BC mass concentration at night was similar to that during the day, both showing an overall increasing trend from 2016 to 2019, with annual rates of increase of 49.0% and 111.7%, respectively.

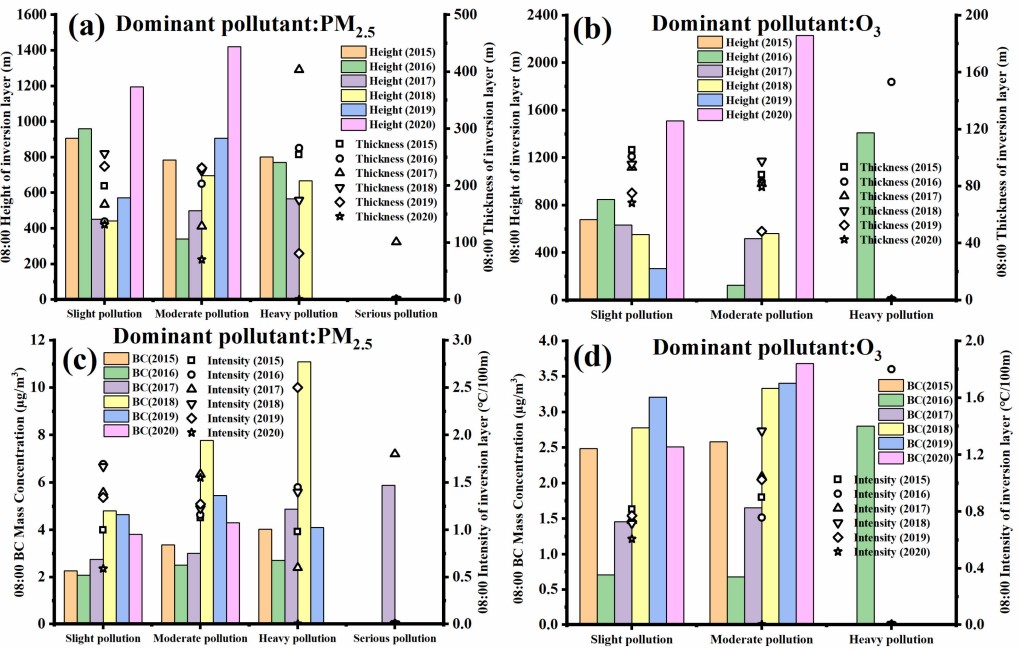

**Figure 5.** (**a**) Annual variations in inversion layer height and inversion layer thickness at 08:00 for the dominant pollutant $PM_{2.5}$ at different air quality levels in 2015–2020; (**b**) Annual variations in inversion layer height and inversion layer thickness at 08:00 for the dominant pollutant $O_3$ at different air quality levels in 2015–2020; (**c**) Annual variations in BC mass concentration and inversion layer intensity at 08:00 for the dominant pollutant $PM_{2.5}$ at different air quality levels in 2015–2020; (**d**) Annual variations in BC mass concentration and inversion layer intensity at 08:00 for the dominant pollutant $O_3$ at different air quality levels in 2015–2020.

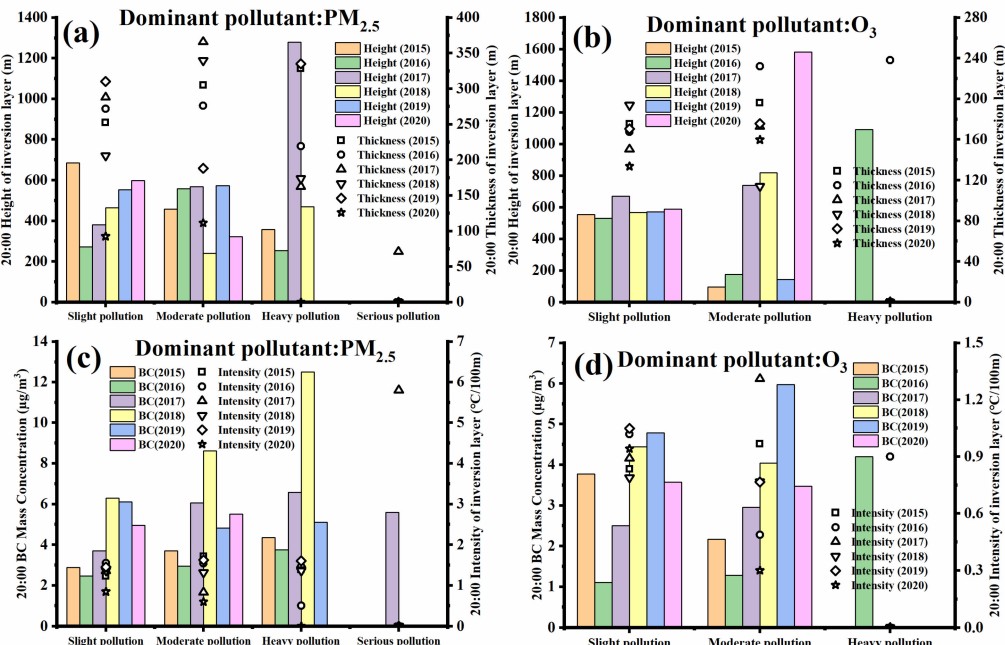

**Figure 6.** (**a**) Annual variations in inversion layer height and inversion layer thickness at 20:00 for the

dominant pollutant $PM_{2.5}$ at different air quality levels in 2015–2020; (**b**) Annual variations in inversion layer height and inversion layer thickness at 20:00 for the dominant pollutant $O_3$ at different air quality levels in 2015–2020; (**c**) Annual variations in BC mass concentration and inversion layer intensity at 20:00 for the dominant pollutant $PM_{2.5}$ at different air quality levels in 2015–2020; (**d**) Annual variations in BC mass concentration and inversion layer intensity at 20:00 for the dominant pollutant $O_3$ at different air quality levels in 2015–2020.

In Figures 5c and 6c, it can be seen that when the air quality was under moderate pollution and the dominant pollutant was $PM_{2.5}$, the BC mass concentration reached large values of 7.8 $\mu g \cdot m^{-3}$ and 8.6 $\mu g \cdot m^{-3}$ in both the daytime and nighttime in 2018. It can be seen in Figures 5 and 6 that the inversion thickness in 2018 was relatively large, at 226.1 m and 340.0 m, which is more conducive to the accumulation of BC. In addition, according to the calculation, the correlation coefficient between the inversion layer height and the BC mass concentration at night in 2015–2020 was $-0.68$, whereas the correlation between the BC mass concentration at both the daytime and nighttime and the inversion intensity was not significant, which indicates that the lower inversion layer height makes the pollutants difficult to diffuse and increases the BC mass concentration at night.

In Figures 5d and 6d, it can be seen that when the air quality was under moderate pollution and when the dominant pollutant was $O_3$, the daytime BC mass concentration decreased significantly in 2015–2016, followed by a yearly upward trend in 2016–2020, with an annual rate of increase of 111.2%; the nighttime BC mass concentration showed an overall upward trend in 2015–2019, with an annual rate of increase of 44.0%, followed by a significant decrease in 2019–2020. In combination with Figures 5b and 6b, it was found that the diurnal and annual changes in the inversion layer height, inversion thickness, and inversion intensity fluctuated greatly. After calculating the correlation with the diurnal and annual changes in BC, it was found that the correlation was not significant and may be greatly affected by anthropogenic emissions.

In Figures 5c and 6c, it can be seen that when the air quality was under heavy pollution and the dominant pollutant was $PM_{2.5}$, the annual variation in the BC mass concentration of the inversion layer thickness during both the daytime and nighttime showed an increasing trend from 2015 to 2018, with annual rates of increase of 59.2% and 62.7% respectively. Then, they both decreased significantly from 2018 to 2019 to 63.1% and 59.2%, respectively.

It can be seen in Figures 5a and 6a that the diurnal and annual changes in the BC mass concentration were not significantly correlated with the inversion layer height, inversion thickness, and inversion intensity. Under severe pollution, it was mainly concentrated in winter, which could have been greatly affected by the long-distance input of foreign pollutants through the cold front.

In general, when the dominant pollutant was $PM_{2.5}$, the annual changes in BC under slight and moderate pollution were greatly affected by the inversion layer height but had no obvious correlation with the inversion thickness and inversion intensity. If the pollution were more serious, BC could be affected by the transmission of overseas pollutants. When the dominant pollutant was $O_3$, the annual changes in BC were less affected by the inversion layer under both slight and moderate pollution and may have been more affected by human activities. In Figures 7c and 8c, it can be seen that when the air quality was under slight pollution and the dominant pollutant was $PM_{2.5}$, the seasonal variation in the daytime BC mass concentration was winter (3.2 $\mu g \cdot m^{-3}$) > summer (2.8 $\mu g \cdot m^{-3}$) > autumn (2.5 $\mu g \cdot m^{-3}$) > spring (2.4 $\mu g \cdot m^{-3}$). Figure 7a shows that the height of the inversion layer during the daytime in spring was 1174.3 m, which was significantly higher than in the other seasons and facilitates the diffusion of pollutants and lowers the BC concentration; the seasonal variation in the BC mass concentration at night was winter (4.0 $\mu g \cdot m^{-3}$) > autumn (3.5 $\mu g \cdot m^{-3}$) > spring (3.2 $\mu g \cdot m^{-3}$) > summer (2.7 $\mu g \cdot m^{-3}$). The inversion layer height at night in the summer was 660.0 m according to Figure 8a and its thickness and intensity were correspondingly 157.0 m and 0.7 $°C/100$ m, which are both noticeably lower compared to the other seasons. Therefore, the inversion layer was weak and thin at night in summer, making it easily destroyed. As a result, the BC mass concentrations were low and rapidly diffused during summer.

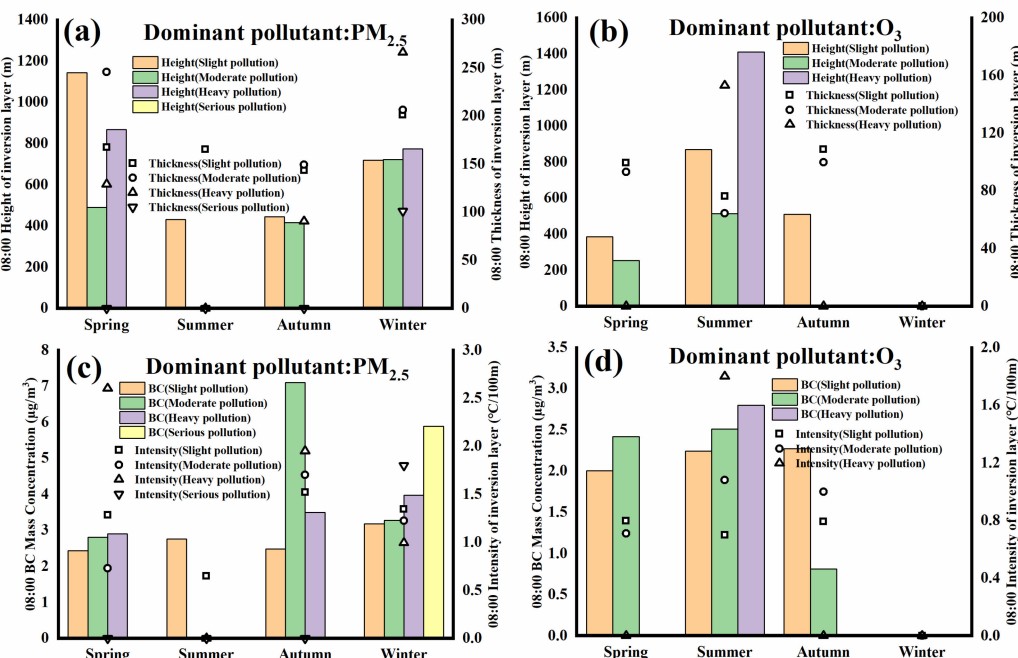

**Figure 7.** (**a**) Seasonal variations in inversion layer height and inversion layer thickness at 08:00 for the dominant pollutant PM$_{2.5}$ at different air quality levels in 2015–2020; (**b**) Seasonal variations in inversion layer height and inversion layer thickness at 08:00 for the dominant pollutant O$_3$ at different air quality levels in 2015–2020; (**c**) Seasonal variations in BC mass concentration and inversion layer intensity at 08:00 for the dominant pollutant PM$_{2.5}$ at different air quality levels in 2015–2020; (**d**) Seasonal variations in BC mass concentration and inversion layer intensity at 08:00 for the dominant pollutant O$_3$ at different air quality levels in 2015–2020.

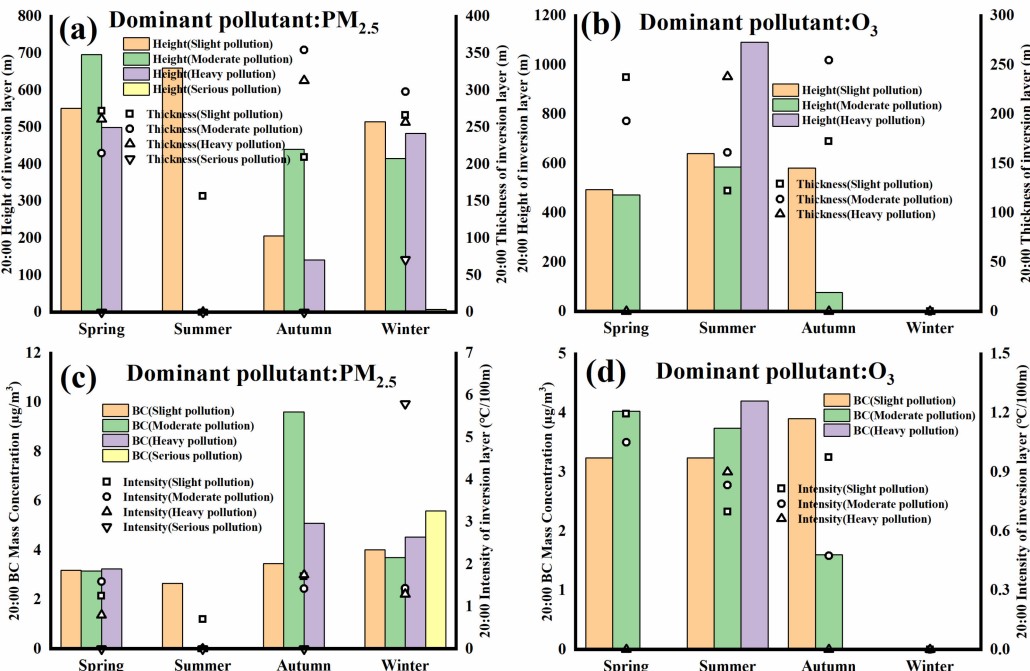

**Figure 8.** (**a**) Seasonal variations in inversion layer height and inversion layer thickness at 20:00 for the dominant pollutant PM$_{2.5}$ at different air quality levels in 2015–2020; (**b**) Seasonal variations in inversion layer height and inversion layer thickness at 20:00 for the dominant pollutant O$_3$ at different

air quality levels in 2015–2020; (**c**) Seasonal variations in BC mass concentration and inversion layer intensity at 20:00 for the dominant pollutant $PM_{2.5}$ at different air quality levels in 2015–2020; (**d**) Seasonal variations in BC mass concentration and inversion layer intensity at 20:00 for the dominant pollutant $O_3$ at different air quality levels in 2015–2020.

In Figures 7d and 8d, it can be seen that when the air quality was under slight pollution and the dominant pollutant was $O_3$, the daytime BC mass concentration was not significantly different in spring, summer, and winter, with a mean value of 2.2 $\mu g \cdot m^{-3}$; the nighttime BC mass concentration was relatively high in autumn at 3.9 $\mu g \cdot m^{-3}$, which was 20.4% and 20.3% higher than that in spring and summer, respectively. Both diurnal BC mass concentrations were not significantly correlated with the inversion layer height, inversion layer thickness, and inversion layer intensity.

In Figures 7c and 8c, it can be seen that when the air quality was under moderate pollution and the dominant pollutant was $PM_{2.5}$, the daytime BC mass concentration was 7.1 $\mu g \cdot m^{-3}$ in autumn, which was 153.6% and 116.5% higher than that in spring and winter, respectively. The daytime inversion layer intensity was 1.7 °C/100m in autumn, which was significantly higher than that in spring and winter, which were 133.5% and 39.3% higher, respectively This indicates that the lower daytime inversion layer height and more vigorous inversion layer intensity in autumn led to higher daytime BC mass concentrations in autumn; the nighttime BC mass concentration was 9.6 $\mu g \cdot m^{-3}$ in autumn, which was 204.8% and 160.2% higher than that in spring and winter, respectively. The inversion layer thickness in autumn was 354.3 m, which was significantly greater than that in spring and winter, 64.7% and 19.2% higher, respectively. The higher inversion layer thickness made the inversion more stable and less susceptible to disruption, which led to the accumulation of BC and increased concentrations.

In Figures 7d and 8d, it can be seen that when the air quality was under moderate pollution and the dominant pollutant was $O_3$, the daytime and nighttime BC mass concentrations were 0.81 $\mu g \cdot m^{-3}$ and 1.6 $\mu g \cdot m^{-3}$ in autumn, respectively, which were significantly lower than those in spring and summer. In combination with Figure 8b, it can be seen that the height of the inversion layer was lowest at night in autumn. Still, the concentration of BC at night in autumn was significantly lower than that in spring and summer, which shows that the thickness of the inversion was significantly stronger in autumn than in spring and summer. This indicates that the mechanism of the inversion stratification on the concentration of atmospheric pollutants was complex. The height, thickness, and intensity of the inversion in different seasons under different dominant pollutants have different influences on the pollutants.

In Figures 7c and 8c, it can be seen that when the air quality was under heavy pollution and the dominant pollutant was $PM_{2.5}$, the seasonal variation in the daytime BC mass concentration was winter (4.0 $\mu g \cdot m^{-3}$) > autumn (3.5 $\mu g \cdot m^{-3}$) > spring (2.9 $\mu g \cdot m^{-3}$), and in Figure 7a, it can be seen that the thickness of the inversion layer was as high as 266.1 m during winter, which was significantly greater than that in the other seasons. Combined with the relatively lower inversion layer height, BC was not quickly diffused. Thus, the daytime BC mass concentration was higher in winter; the seasonal variation in the nighttime BC mass concentration was autumn (5.1 $\mu g \cdot m^{-3}$) > winter (4.5 $\mu g \cdot m^{-3}$) > spring (3.3 $\mu g \cdot m^{-3}$), as shown in Figure 8a, and the nighttime inversion layer height was 141.0 m in autumn, which was lower than those in spring and winter by 71.8% and 70.8%, respectively. The lower inversion layer height and the strong and thick inversion laminar junction at night in autumn likely led to the accumulation of pollutants, making the BC mass concentration higher at night in autumn.

In general, when the dominant pollutant was $PM_{2.5}$, the diurnal variation in the BC seasonal variation under slight pollution, moderate pollution, and severe pollution was significantly affected by the inversion layer. When the dominant pollutant was $O_3$, the seasonal change in BC had no obvious correlation with the inversion layer, which was the same as the annual change in BC.

## 4. Conclusions

There was a clear trend toward higher air quality from 2015 to 2020, with the number of days under excellent air quality reaching 96 in 2020, up from only 33 in 2015, showing an annual increase of 38.2%. The number of days with $PM_{2.5}$ as the dominant pollutant decreased yearly from 89 in 2015 to 18 in 2020, representing an annual decrease of 16.0%. The number of days with $O_3$ as the dominant pollutant rose, with 54 in 2015 and the highest number of days in 6 years at 78 in 2019, an annual rate of increase of 11.1%.

When the dominant pollutant was $PM_{2.5}$, the annual variation in the BC mass concentration was the same under slight pollution, moderate pollution, and heavy pollution, all decreasing in the middle of 2015–2016 and then increasing yearly in 2016–2018, with annual rates of increase of 73.8%, 105.5%, and 156.3% respectively, and after reaching the maximum in 2018 with 9.6%, 20.6%, and 62.1%, the annual rates of decline began to drop from 2020. The BC mass concentration was negatively correlated with the inversion layer height under both slight and moderate pollution. The seasonal variation in the BC mass concentration was winter > autumn > summer > spring under slight pollution and autumn > winter > spring under both moderate and heavy pollution, which were mainly influenced by the difference between the inversion layer junction and wind speed. In general, when the dominant pollutant was $PM_{2.5}$, the annual and seasonal changes in the BC mass concentration were greatly affected by the inversion layer, and the seasonal changes were additionally affected by the wind speed.

When the dominant pollutant was $O_3$, the annual change in the BC mass concentration was the same under slight pollution and moderate pollution, both of which decreased significantly in 2015–2016, being 70.6% and 60.0% lower in 2016 than in 2015, respectively, and then both showed an increasing yearly trend in 2016–2019, with annual rates of increase of 112.2% and 138.6%, respectively. After reaching a maximum in 2019 and then beginning to decrease, both were significantly negatively correlated with the wind speed, with correlation coefficients of −0.79 and −0.68, respectively. The seasonal changes in the BC mass concentration were mainly influenced by the wind speed in autumn > summer > spring under slight pollution and spring > summer > autumn under moderate pollution. In general, when the dominant pollutant was $O_3$, the annual and seasonal changes in the BC mass concentration were only affected by the wind speed.

This study has comprehensively analyzed BC aerosol in Nanjing from 2015 to 2020, revealed different pollution characteristics and influence factors of BC aerosol under $PM_{2.5}$ pollution and $O_3$ pollution, respectively, and made certain achievements. However, there are still some shortcomings and areas worthy of improvement. This study only focuses on the analysis of the concentration and temporal and seasonal variation characteristics of black carbon aerosol in Nanjing but lacks an in-depth analysis of its formation mechanism, which needs to be combined with models and other chemical composition analyses. In the future, the source of black carbon aerosol can be further analyzed and quantified using inorganic ion composition, the isotope analysis method, and the PMF model. At the same time, this study only analyzes the data from one observation point in Nanjing. In the future, it is hoped that the spatial distribution of BC can be more comprehensively and deeply discussed by conducting multi-point observation experiments.

**Author Contributions:** Conceptualization, Y.P., H.W. and S.S.; Investigation, Y.P., H.W. and Y.T.; Methodology, W.L. and Y.T.; Visualization, Y.P. and H.W.; Writing—original draft, Y.P. and H.W.; Writing—review and editing, Y.P., H.W., B.Z. and T.Z. All authors have read and agreed to the published version of the manuscript.

**Funding:** This research was funded by the Natural Science Foundation of China (NSFC) research project (Grant No. 42192512, No. 41830965, and No. 41805096) and the Postgraduate Research and Practice Innovation Program of Jiangsu Province in 2022 (Grant No. SJCX22_0361).

**Institutional Review Board Statement:** The authors declare that they have no known competing financial interests or personal relationships that could have appeared to influence the work reported in this paper.

**Informed Consent Statement:** Not applicable.

**Data Availability Statement:** The air quality levels used in this paper were obtained from the historical data from China's air quality online monitoring and analysis platform. Available online: https://www.aqistudy.cn/historydata/ (accessed on 9 August 2021). The atmospheric boundary layer data were from daily soundings at 08:00 and 20:00 from the University of Wyoming weather data website. Available online: http://www.weather.uwyo.edu/upperair/sounding.html (accessed on 9 August 2021).

**Conflicts of Interest:** The authors declare no conflict of interest.

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
