# Peer review of "Analysis of BC Pollution Characteristics under PM2.5 and O3 Pollution Conditions in Nanjing from 2015 to 2020"

_atmosphere, doi:10.3390/atmos13091440_

Round 1
Reviewer 1 Report
This article discussed the pollution under PM2.5 and O3 pollution in Nanjing, paying more attention to the yearly average also Seasonal and diurnal variation, but the further investigation seems to lack evidence, the author should provide more proof to convince the reader with a clear discussion, it should be a major revision before acceptance.
L54: pollution stage where aerosols and O3 coexist, these two pollutions always exist, but we are talking about the complex mechanism of how to control their generation.
L76-79: the relationship also could refer to PBL impact refer to the following paper:
Su, T., Li, Z., Li, C., Li, J., Han, W., Shen, C., Tan, W., Wei, J., and Guo, J.: The significant impact of aerosol vertical structure on lower atmosphere stability and its critical role in aerosol–planetary boundary layer (PBL) interactions, Atmos. Chem. Phys., 20, 3713–3724, https://doi.org/10.5194/acp-20-3713-2020, 2020.
L95: “Its influencing factors can help promote the This will help to promote systematic observation of BC aerosols” this sentence led to misunderstanding.
L113:a site location and brief introduction should be added to replace this paragraph.
L120: the so-called “cutter head” is not an appropriate word.
L141: where is the sounding site data used in this manuscript?
L145: “Figure 1(a) shows that the annual changes in the number of days with different air quality levels in Nanjing from 2015 to 2020 are significant. ” should be rewritten.
L147-156: the excellent and another level corresponding to how many PM2.5, and O3 should be provided, such as excellent(PM2.5<5 μg·m-3) or make a standard number list.
L191: “Blue Sky Defence Action Plan from 2018 to 2020.” Reference or links need to add.
L223: Fig.2 only divides the primary pollutant and analysis the change just similar to Fig.1
L226: “BC when the primary pollutant is PM2.5” how to explain?
L313-314: “The inverse thermocline within the boundary layer is similar to a "dome," which prevents the diffusion of pollutants to higher altitudes, causing a significant accumulation of pollutants in the area below the inverse thermocline.” Just like Ding et al have proposed this phenomenon.
Ding, A.J.; Huang, X.; Nie, W.; Sun, J.N.; Kerminen, V.M.; Petaja, T.; Su, H.; Cheng, Y.F.; Yang, X.Q.; Wang, M.H.; et al. Enhanced 495 haze pollution by black carbon in megacities in China. Geophys. Res. Lett. 2016, 43, 2873-2879.
L329-438: From Fig. 5 to Fig. 8, there are too many annual variations or seasonal variations illustrated in the manuscript, the author just creates one plot model and repeats change to other pollutant or inversion layer height and another factor, but the key scientific issue has not been fully discussed, suggest to further discuss the difference between each other or use the boxplot to compare the Seasonal variations.
L439: the conclusion needs to summarize the all founding, not just copy some results to this part.

Author Response
Response to Reviewer
This article discussed the pollution under PM2.5 and O3 pollution in Nanjing, paying more attention to the yearly average also Seasonal and diurnal variation, but the further investigation seems to lack evidence, the author should provide more proof to convince the reader with a clear discussion, it should be a major revision before acceptance.
Response: I would like to take this great opportunity to thank you for the valuable comments. We have provided more evidence to clearly convince readers to discuss and seriously revise all the suggestions you have made. I believe that the addressing of these comments has greatly improved the quality of this manuscript.
Point 1: L54: pollution stage where aerosols and O3 coexist, these two pollutions always exist, but we are talking about the complex mechanism of how to control their generation.
Response 1: Thank you very much for your comments. We have revised this sentence based on your comments, as detailed in L64.
Point 2: L76-79: the relationship also could refer to PBL impact refer to the following paper:
Su, T., Li, Z., Li, C., Li, J., Han, W., Shen, C., Tan, W., Wei, J., and Guo, J.: The significant impact of aerosol vertical structure on lower atmosphere stability and its critical role in aerosol–planetary boundary layer (PBL) interactions, Atmos. Chem. Phys., 20, 3713–3724, https://doi.org/10.5194/acp-20-3713-2020, 2020.
Response 2: Thank you very much for your comments. We have cited the paper you provided us with, as detailed in L98-102.
Point 3: L95: “Its influencing factors can help promote the This will help to promote systematic observation of BC aerosols” this sentence led to misunderstanding.
Response 3: Thank you very much for your comments. Our original writing was indeed misleading, and we have rewritten the phrase, as detailed in L127-131.
Point 4: L113:a site location and brief introduction should be added to replace this paragraph.
Response 4: Thank you very much for your comments. We have added a site location and a brief description to replace that paragraph, as detailed in L134-149.
Point 5: L120: the so-called “cutter head” is not an appropriate word.
Response 5: Thank you very much for your comments. We have replaced "cutting head" with "cutting joint", as detailed in L156.
Point 6: L141: where is the sounding site data used in this manuscript?
Response 6: Thank you very much for your comments. The meteorological element data of our observation station is from the atmospheric observation base of Nanjing University of information technology. This data has no public access channel and needs to be downloaded from the station. We will supplement the data as an attachment later.
Point 7: L145: “Figure 1(a) shows that the annual changes in the number of days with different air quality levels in Nanjing from 2015 to 2020 are significant. ” should be rewritten.
Response 7: Thank you very much for your comments. We have rewritten this sentence, as detailed in L180.
Point 8: L147-156: the excellent and another level corresponding to how many PM2.5, and O3 should be provided, such as excellent(PM2.5<5 μg·m-3) or make a standard number list.
Response 8: Thank you very much for your comments. We have developed the table of air quality levels corresponding to PM2.5 and O3, as detailed in L200.
Point 9: L191: “Blue Sky Defence Action Plan from 2018 to 2020.” Reference or links need to add.
Response 9: Thank you very much for your comments. We have added a link to “Three-Year Action Plan for the Defense of the Blue Sky implemented from 2018 to 2020” (http://zrzy.tianshui. gov.cn/dayin-c9c80aba59ed471fa05f7eddd55abeea.htm), as detailed in L231.
Point 10: L223: Fig.2 only divides the primary pollutant and analysis the change just similar to Fig.1
Response 10: Thank you very much for your comments. Our Figure 2 removes the excellent level compared to Figure 1, and pays more attention to the annual and seasonal changes under different primary pollutants in air pollution. Figure 1 briefly depicts the overall annual and seasonal variations of air quality in Nanjing from 2015 to 2020, while Figure 2 focuses more on the differences in air pollution characteristics under different pollutants and the corresponding differences in influencing factors, which further leads to a more specific analysis of the differences in BC pollution characteristics under different primary pollutants from meteorological elements and the inversion layer in the later section.
Point 11: L226: “BC when the primary pollutant is PM2.5” how to explain?
Response 11: Thank you very much for your comments. We are very sorry for the problem with the organization of our language and have reworked this sentence, as detailed in L269.
Point 12: L313-314: “The inverse thermocline within the boundary layer is similar to a "dome," which prevents the diffusion of pollutants to higher altitudes, causing a significant accumulation of pollutants in the area below the inverse thermocline.” Just like Ding et al have proposed this phenomenon.
Ding, A.J.; Huang, X.; Nie, W.; Sun, J.N.; Kerminen, V.M.; Petaja, T.; Su, H.; Cheng, Y.F.; Yang, X.Q.; Wang, M.H.; et al. Enhanced 495 haze pollution by black carbon in megacities in China. Geophys. Res. Lett. 2016, 43, 2873-2879.
Response 12: Thank you very much for your comments. We have added citation marks as detailed in L367.
Point 13: L329-438: From Fig. 5 to Fig. 8, there are too many annual variations or seasonal variations illustrated in the manuscript, the author just creates one plot model and repeats change to other pollutant or inversion layer height and another factor, but the key scientific issue has not been fully discussed, suggest to further discuss the difference between each other or use the boxplot to compare the Seasonal variations.
Response 13: Thank you very much for your comments, Thank you very much for the reviewer's comments on our article, it is very important, due to your suggestions we found the shortcomings in the article, we have added a summary of the discussion of the differences in the article detailed in L363-518, mainly illustrating the differences in the factors affecting BC under different primary pollution, we will continue to improve our scientific research in the future in accordance with your suggestions, in-depth research and achieve more results!
Point 14: L439: the conclusion needs to summarize the all founding, not just copy some results to this part.
Response 14: Thank you very much for your comments. We have revised our conclusions to add a more detailed summary, as detailed in L519-561.
We look forward to hearing from you regarding our submission. We would be glad to respond to any further questions and comments that you may have.

Reviewer 2 Report
In general, the paper is well written and has scientific value.
I have one general remark - The Authors are describing BC in the first paragraph extensively and multidisciplinary while in the next section they are referring only to a specific region. It is of course not a disadvantage, but I would like to see a deeper discussion about the air pollution problem and similar long-term studies in China and around the world. Especially for places with similar seasons and source characteristics. For example you can see the latest research analyzing solid fuels heating and data from the last 10 years in the most polluted city in Europe (Krakow). It shows the similar observations as yours related to air quality (for reference check the special issue Sensors for Air Quality Monitoring of the MDPI SENSORS journal 2021). There are also other studies from S America as well.
Similar comment to the discussion part.
Please correct the legend in Figure 8a
Author Response
Response to Reviewer
Thank you very much for your recognition of our manuscript. I would like to take this great opportunity to thank you for the valuable comments. We have addressed all the comments carefully, and the revised portions are highlighted in red in the manuscript. I believe that the addressing of these comments has greatly improved the quality of this manuscript.
Point 1: I have one general remark - The Authors are describing BC in the first paragraph extensively and multidisciplinary while in the next section they are referring only to a specific region. It is of course not a disadvantage, but I would like to see a deeper discussion about the air pollution problem and similar long-term studies in China and around the world. Especially for places with similar seasons and source characteristics. For example you can see the latest research analyzing solid fuels heating and data from the last 10 years in the most polluted city in Europe (Krakow). It shows the similar observations as yours related to air quality (for reference check the special issue Sensors for Air Quality Monitoring of the MDPI SENSORS journal 2021). There are also other studies from S America as well.
Response 1: Thank you very much to the reviewers for your comments on our article, your suggestions are very important and we think that air pollution is something that needs to be discussed in depth and studies from all over the world should be studied carefully. We have carefully read the literature you provided us with, which shows observations similar to our air quality and also points out the main sources of pollution, and other literature we have reviewed, which are very informative. Our article may lack a more in-depth discussion of the sources of air pollution and the study is mainly focused on the context of compound pollution in China, so we have added two new relevant papers on other regions of China, as detailed in L65-75.
Point 2: Similar comment to the discussion part.
Response 2: Thank you very much for your comments. As you said our talk section also lacks some deeper studies and is limited to Nanjing, we hope that in the future we can analyze the sources in depth and conduct multi-site observations to carry out more in-depth regional pollution studies. Thanks to your suggestions, I found the shortcomings in my current work, and I will follow your suggestions to improve my research and achieve more in the future.
Point 3: Please correct the legend in Figure 8a
Response 3: Thank you very much for your comments. We have carefully checked the legend and raw data in Figure 8(a) that there is no problem. Severe pollution only occurs on 1 day in winter and the height of the inversion layer at 20:00 when severe pollution occurs is only 7 m, so it is not easy to be seen.
We look forward to hearing from you regarding our submission. We would be glad to respond to any further questions and comments that you may have.

Reviewer 3 Report
This subject addressed is within the scope of the journal. However, the manuscript in the present version contains several problems. Appropriate revisions should be undertaken in order to justify recommendation for publication.
1. It is mentioned that the air quality levels and pollutant observations used in this paper were obtained from the http://www.cnemc.cn/ (accessed on 9 August 2022). However, on cross verification, data not found in accessible form on relevant website. The data should be reported as a supplementary file (inputs and output).and kindly explain how possible that you access data on 9th august and within a day analysis everything completed and submit manuscript on 10th August?
2. For readers to quickly catch your contribution, it would be better to highlight major difficulties and challenges, and your original achievements to overcome them, in a clearer way in abstract and introduction.
3. There is a serious concern regarding the novelty of this work. What new has been proposed?
4. Abstract needs to modify and to be revised to be quantitative. You can absorb readers' consideration by having some numerical results in this section.
5. There are some occasional grammatical problems within the text. It may need the attention of someone fluent in English language to enhance the readability.
6. The discussion section in the present form is relatively weak and should be strengthened with more details and justifications.
7. In conclusion section, limitations and recommendations of this research should be highlighted.
8. The authors have to add the state-of-the art references in the manuscripts.
9. It is mentioned that Nanjing is adopted as the case study. What are other feasible alternatives? What are the advantages of adopting this case study over others in this case? How will this affect the results? The authors should provide more details on this.
Author Response
Response to Reviewer
This subject addressed is within the scope of the journal. However, the manuscript in the present version contains several problems. Appropriate revisions should be undertaken in order to justify recommendation for publication.
Response: I would like to take this great opportunity to thank you for the valuable comments. We have addressed all the comments carefully, and the revised portions are highlighted in red in the manuscript. I believe that the addressing of these comments has greatly improved the quality of this manuscript and justify recommendation for publication.
Point 1: It is mentioned that the air quality levels and pollutant observations used in this paper were obtained from the http://www.cnemc.cn/ (accessed on 9 August 2022). However, on cross verification, data not found in accessible form on relevant website. The data should be reported as a supplementary file (inputs and output).and kindly explain how possible that you access data on 9th august and within a day analysis everything completed and submit manuscript on 10th August?
Response 1: Thank you very much for your comments. We have updated the data site (https://www.aqistudy.cn/historydata/) for you to access and we will upload the data as a supplemental file afterwards. Secondly, we apologize that you misunderstood that we would access the data on August 9 and complete and submit the manuscript within one day. We intended that the link to that data site on August 9 was valid and accessible, not invalid and unopenable, and apologize for that.
Point 2: For readers to quickly catch your contribution, it would be better to highlight major difficulties and challenges, and your original achievements to overcome them, in a clearer way in abstract and introduction.
Response 2: Thank you very much for your comments. We have included our contribution in the abstract, as detailed in L34-37. Also we have rewritten the last part of the introduction to highlight the main difficulties we had to overcome, as well as the achievements and contributions made, as detailed in L114-131
Point 3: There is a serious concern regarding the novelty of this work. What new has been proposed?
Response 3: Thank you very much for your comments. Our manuscript first summarizes the previous studies and finds that many scholars have studied about the pollution characteristics of BC under short- and medium-term air pollution and analyzed them with meteorological conditions. However, in the context of composite atmospheric pollution, there are few studies that divide air pollution into air pollution under different primary pollutants and then further explore the characteristics of BC pollution under long time series, and the interrelationship between the characteristics of black carbon aerosol changes and boundary layer characteristics under different years and seasons as well as the differences in the influence of its meteorological elements need to be further analyzed and explored. In this paper, we firstly divide the air pollution in Nanjing area from 2015 to 2020 into those when the primary pollutant is PM2.5 and those when the primary pollutant is O3, and then conduct a study on the characteristics of BC pollution under two different primary pollutants. And we also discuss the influence of boundary layer characteristics on BC pollution characteristics, while combining meteorological elements can more comprehensively analyze the influencing factors of BC pollution characteristics under different primary pollutants. The results show that the BC pollution characteristics and its influencing factors are different under different primary pollutants. The study of BC evolution in air pollution under different primary pollutants is important to further improve the capability and level of global climate change research and prediction, and can provide a scientific basis for assessing environmental health and climate impacts.
Point 4: Abstract needs to modify and to be revised to be quantitative. You can absorb readers' consideration by having some numerical results in this section.
Response 4: Thank you very much for your comments. We have modified the abstract to provide some numerical results to attract readers, as detailed in L14-28.
Point 5: There are some occasional grammatical problems within the text. It may need the attention of someone fluent in English language to enhance the readability.
Response 5: Thank you very much for your comments. We have asked someone who is fluent in spoken English to touch up the phrasing of our articles and fix some grammatical problems in the articles.
Point 6: The discussion section in the present form is relatively weak and should be strengthened with more details and justifications.
Response 6: Thank you very much for your comments. We have strengthened the discussion and provided more details and reasons, as detailed in L179-518.
Point 7: In conclusion section, limitations and recommendations of this research should be highlighted.
Response 7: Thank you very much for your comments. We have added the limitations and suggestions of this study in the conclusion part, as detailed in L550-561.
Point 8: The authors have to add the state-of-the art references in the manuscripts.
Response 8: Thank you very much for your comments. We have added some of the advanced literature to the article, as detailed in L58,65-75.
Point 9: It is mentioned that Nanjing is adopted as the case study. What are other feasible alternatives? What are the advantages of adopting this case study over others in this case? How will this affect the results? The authors should provide more details on this.
Response 9: Thank you very much for your comments. Except Nanjing, which can be used as a case study, other cities with compound pollution can be used as case studies, such as Beijing. As one of the fastest growing economic regions in the world, the Yangtze River Delta region has high BC emissions, which have a large impact on regional climate and air quality. Nanjing, as the central city in the Yangtze River Delta and East China, has the typical characteristics of compound pollution in its atmosphere. The difference between the number of days of PM2.5 pollution and the number of days of O3 pollution in a year is relatively small in Nanjing compared to other cities, especially from 2015 onwards, and the total proportion of 2 air pollution in all air pollution days from 2015 to 2020 is more than 94%, indicating that Nanjing is well suited to divide air pollution into PM2.5 pollution and O3 pollution to be discussed separately. Nanjing is more typical and the results are more accurate compared to other cases, which makes out the pollution characteristics of BC under PM2.5 and O3 pollution conditions and its influencing factors more reliable. It is important to further improve the capability and level of global climate change research and prediction. Meanwhile, it can better provide scientific basis for the prevention and control of regional air pollution measures, assessment of environmental health and climate impact.
We look forward to hearing from you regarding our submission. We would be glad to respond to any further questions and comments that you may have.

Round 2
Reviewer 1 Report
All of my concens has been modified in the revison, this manuscript could be accept in principle.
Reviewer 3 Report
Accept in present form